# The Scenic Beauty of Geosites and Its Relation to Their Scientific Value and Geoscience Knowledge of Tourists: A Case Study from Southeastern Spain

**Getaneh Addis Tessema** [1,2,*] , **Jean Poesen** [1,3] , **Gert Verstraeten** [1] , **Anton Van Rompaey** [1] and **Jan van der Borg** [1]

1 Division of Geography and Tourism, Department of Earth and Environmental Sciences, KU Leuven, Celestijnenlaan 200E, 3001 Leuven, Belgium; jean.poesen@kuleuven.be (J.P.); gert.verstraeten@kuleuven.be (G.V.); anton.vanrompaey@kuleuven.be (A.V.R.); jan.vanderborg@kuleuven.be (J.v.d.B.)
2 Department of Tourism and Hotel Management, Bahir Dar University, Bahir Dar P.O. Box 79, Ethiopia
3 Faculty of Earth Sciences and Spatial Management, Maria Curie-Sklodowska University, Kraśnicka 2cd, 20-718 Lublin, Poland
* Correspondence: getanehaddis.tessema@kuleuven.be or gechzadd23@gmail.com

**Abstract:** Scenic beauty is one of the most-commonly used indicators in the inventory and assessment of geosites for geoconservation, geoheritage management and geotourism development. It is an important driver of tourists to visit natural areas and it also provides support for the protection of natural heritage. Previous studies on scenic beauty mainly focused on landscape preference and physical characteristics of geosites that affect scenic beauty appreciation. The relationships between the scenic beauty of geosites, their scientific value and the geoscience knowledge of tourists has not been empirically investigated in detail. Hence, this study investigates this relationship using 34 geosites from southeastern Spain. For this purpose, 29 respondents with a geoscience background and who all visited the 34 geosites, 43 respondents with a geoscience background but who did not visit the geosites, and 104 respondents with no geoscience background and who did not visit the geosites, participated in a survey. The first group rated the scenic beauty and the scientific value of the geosites based on a direct field visit during which the scientific background of these geosites was given. On the other hand, the latter two groups rated scenic beauty using representative photos of the geosites. A five-point Likert scale was used to rate the scenic beauty and the scientific value of the geosites. We found a significant relationship between the scenic beauty of geosites and their scientific value, and this relationship becomes more significant if the geoscientific knowledge of the respondents increases. One-way ANOVA results indicated that a geoscience background contributed to higher perceived scenic beauty, especially for those geosites that in general were considered as more scenic by all the respondent groups. It was also found that geosites with viewpoints received in general higher scenic beauty and scientific value ratings.

**Keywords:** assessment; geo-interpretation; geosite value; geosite cluster; geotourism

## 1. Introduction

Geodiversity, which has been recognized as a concept worth investigating from the 1990s onwards, is in recent times more frequently introduced into scientific nomenclature. It has aroused a strong interest of researchers from geology, geography, biology, spatial planning, general tourism as well as national geotourism and cultural heritage [1]. Geodiversity refers to the "natural range (diversity) of geological (rocks, minerals, fossils), geomorphological (landforms, topography, physical processes), soil (pedological) and hydrological features. It includes their assemblages, structures, systems and contributions to landscapes" ([2], p. 14). Geodiversity is the "abiotic equivalent" or "natural twin" of

the term "biodiversity" [3,4]. It provides an important resource for human development, and it also influences the distribution of flora and fauna and ecosystem functioning [5,6]. In addition to its scientific value, geodiversity is also an important resource for education, tourism and cultural identity of local communities [7].

Geosites are part of geodiversity with a certain value and hence identified as worthy of geoheritage and geoconservation [8]. Geosites are defined as "geological or geomorphological (geodiversity) objects that have acquired a scientific, cultural/historical, aesthetic and/or social/economic values" ([9], p. 440). A geosite can take different forms, including a "landscape, a group of landforms, a single landform, a rock outcrop, a fossil bed or a fossil" ([10], p. 6). Geosites are valuable assets for science and education [11–14] as well as for geotourism development [5,13–16]. For example, a study of gullies in twelve representative gully regions in nine European countries, including Spain, indicated that these geosites offer unique educational lessons about present-day geomorphological processes, stages of historical gully erosion reflecting past human–environment interactions and function as a geological window [17].

Assessing the potential of geosites is necessary for geoconservation, geoheritage management and geotourism development [18], and one of the most common criteria for such purposes is scenic beauty [7,16–23]. Scenic beauty is also used by UNESCO as a criterion to register natural sites in its World Heritage Site list [24]. UNESCO uses "exceptional natural beauty and aesthetic importance" as a criterion to register natural World Heritage Sites.

Scenic beauty contributes to the overall value of nature, providing a reason for its protection and preservation [23–29]. Additionally, beautiful scenery is an important component for tourism [30] and tourists' emotional satisfaction [31]. Furthermore, it has been found that people's happiness is greater in more scenic locations [32], and people living in more scenic environments report better health [33]. As a result, their assessment helps for successful destination development and management [34]. Quantification and empirical studies on scenic beauty shed light and provide support for the management of geosites [35].

There are two approaches to scenic beauty assessment: the objectivist and subjectivist approaches [36]. The objective approach involves scenic beauty to be assessed by experts based on formal knowledge [37], using key elements and features of the geosite [38]. In this approach, to assess scenic beauty, the expert applies certain criteria subjectively presented as objectivity [30] such as the number of viewpoints, surface area, surrounding landscape and nature [22], color diversity and combination, the presence of water and vegetation, absence of human-induced deterioration and proximity to the observed features [19]. On the other hand, the subjectivist approach involves deriving scenic beauty based on people's perceptions and preferences [37].

There is no consensus on the two approaches, and the debate on whether scenic beauty is inherently related to the physical characteristics of geosites or whether it is objective has continued for years [39]. It has been indicated that "some agreement was found regarding landforms most likely to be perceived as scenic or unattractive by experts and non-experts" ([39], p. 1). However, the objective approach is criticized for its inadequate reliability [40]. Lothian ([36], p. 25) argues that, unlike the objectivist method, the subjectivist approach offers a method that is "scientifically and statistically rigorous, is replicable and objective, reflects the preferences of the community and can indicate the degree of accuracy of its results". This approach is dominant in scenic beauty assessment research [37].

The use of photographs for landscape scenic beauty assessment is generally considered acceptable [38,41–45]. Daniel [40] found that visual scenic beauty assessment based on color photographs mostly matches assessments based on on-site experience. As a result, several studies on the scenic beauty of landscapes, have been conducted using photos [39,41,43–47].

Many studies on scenic beauty of geosites mainly relate to landscape preference [26,41,48–56] and such an approach fails to assess people's ratings of the quality of the landscape [57].

An important issue in the assessment of scenic beauty is which factors influence geosite scenic beauty ratings. In previous studies, two factors were identified: biophysical and personal. Among the biophysical factors that positively influence scenic beauty rating were water bodies, naturalness/wilderness, vegetation/forest and color diversity/contrast [46,58]. In addition, landform size and diversity [59], openness and uniqueness [29], shape and scale [60], the presence of mountains/hilly landform and well-preserved man-made features [44,46], number of viewpoints and absence of human deterioration [22] all influence scenic beauty ratings. On the other hand, the influence of personal factors such as age, gender and education on scenic beauty ratings were investigated and some studies found a significant difference in scenic beauty rating based on these factors [38,42,61] while others did not [43,55,62,63].

However, there are other important factors that could have a relationship with scenic beauty, but that were not given due attention in previous studies. One of these is the scientific and educational value of geosites. Similar to scenic beauty, the scientific and educational value of geosites is also one of the most common criteria in the assessment of geosites for geoconservation, geoheritage management and geotourism development [5,7,18–21] and it is also one of UNESCO's criteria to register natural sites in its World Heritage Site list [24]. The fact that scenic beauty, and scientific and educational value are often assessed separately suggests that these are seen as complementary values but without a strong relationship between both. The relation between both values has, to our knowledge, not been quantified [17,20,21]. A better understanding of the relationship between both values is important when selecting, developing, conserving and managing geosites.

The scientific and educational value of geosites is assessed in many ways, and there is no commonly agreed method for using these values in geosite assessments. Some studies assess scientific value (and educational value is not included in their methodology) with its own sub-indicators such as geologic history, rarity, integrity, representativeness, (geo)diversity and scientific knowledge [5,19,21]. Educational value is separately evaluated with its own (sub)indicators [5,7]. On the other hand, Vujičić et al. [22] assessed scientific and educational values together (using four indicators: rarity, representativeness, knowledge of geoscientific issues and level of interpretation). Other researchers included educational value as one sub-indicator of scientific value [64,65]. In our study, scientific and educational values were considered as one value of geosites, similar to Vujičić et al. [22], and will hereafter be called scientific value.

The other important factor that received no to little empirical investigation so far is whether or not geoscience knowledge contributes to a higher perceived scenic beauty rating. There are philosophical arguments about the effects of scientific knowledge on the aesthetic appreciation of the natural environment. For example, Carlson [66] argues that knowledge of the different natural environments and their systems and elements is required to aesthetically appreciate nature. Despite its importance for allowing a complex, deep, and meaningful aesthetic appreciation of nature, criticism arises on Carlson's argument since people also appreciate nature without scientific knowledge [25], and there is nothing wrong with such judgments [67]. Though Stecker admitted that some knowledge can enhance one's appreciation of nature by enabling one to think and perceive nature in more complex ways, he indicated that there are certain very common appreciative responses to nature, such as appreciating a thundering waterfall for its grandeur, which requires less intellect. These are conceptual arguments about the role of scientific knowledge in the aesthetic appreciation of nature. Hence, an empirical study is required to test whether scientific knowledge contributes to higher scenic beauty ratings. As scientific knowledge is a broad concept, geoscience knowledge is used for our case.

The objective of this study is, therefore, to investigate the relationship between the scenic beauty of geosites and their scientific value and geoscience knowledge. Thus, we established the following two hypotheses:

**Hypothesis 1 (H1).** *There is a positive relationship between the scenic beauty assessment by tourists and scientific value of geosites.*

**Hypothesis 2 (H2).** *Geoscience knowledge contributes to a higher perceived scenic beauty.*

These hypotheses were tested in a case study of 34 geosites from southeastern Spain.

## 2. Materials and Methods

### 2.1. Description of the Study Area

With about 27% of the country's territory, Spain has the largest surface of protected natural areas in the European Union ([68], p. 307). It has also some of the best exposed geology in Europe, due to its mountainous nature, extensive coastline and somewhat arid climate [69]. The study area covers three provinces located in southeastern Spain: i.e., Granada, Almeria and Murcia (Figure 1). The region is home to two UNESCO Global Geoparks (Cabo de Gata-Níjar and Granada).

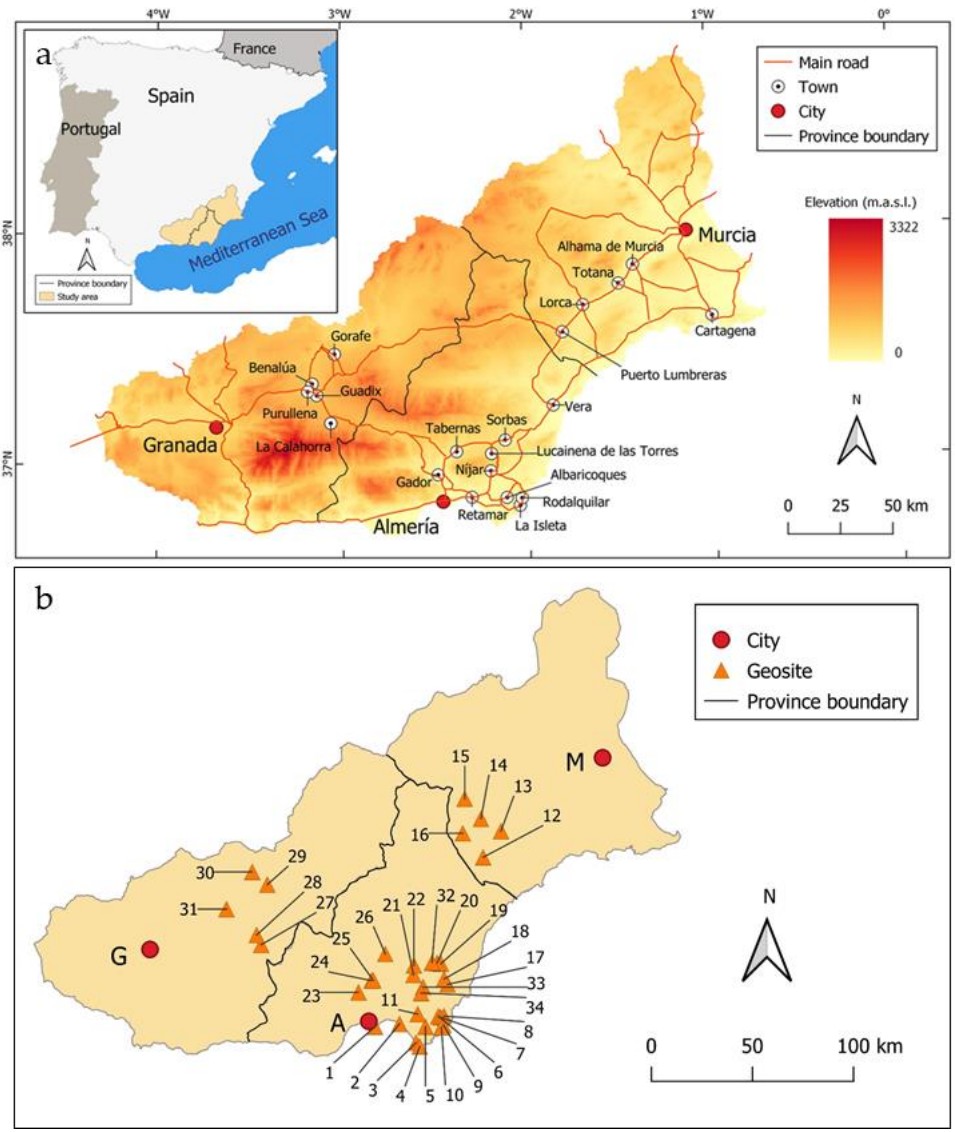

**Figure 1.** (**a**) Map showing the location of the study area in southeast Spain. Cities/towns indicated on the map are those near the geosites. (**b**) Location map of the 34 geosites in Table A1 in Appendix A; for their scenic beauty ratings, see Table A4 in Appendix A; for their scientific value rating, see Table A5 in Appendix A; for their photos, see Questionnaire S1 in the Supplementary Materials.

The study area offers a unique opportunity for teaching field geology and geomorphology [70], and many student field trips are organized by European universities to the region [71,72].

Geomorphological and Geological Setting of the Study Area

The study area is amongst the driest regions of Europe [73] which makes it an ideal place to learn and enjoy about the geology and geomorphology as the vegetation cover is rather limited allowing many geomorphological and geological features to be observed easily [71]. The region is characterized by a series of mountain ranges or Sierras (Betic chain, resulting from the Alpine orogeny and mainly consisting of hard Paleozoic and Mesozoic rocks) and uplifted Cenozoic basins dominated by unconsolidated sediments [74–77]. From a lithological point of view, volcanic, sedimentary and metamorphic rock types outcrop in the area [78–82], resulting in diverse landform types. The region is also tectonically active [74,75,82–84] which allows to observe active faults, horst and graben landforms as well as volcanic features [77,83–86]. Erosional and depositional features include various types of mass movements, gullies, badlands, fans and landforms resulting from fluvial and coastal dynamics [87–89]. Several mineral deposits (e.g., lead, iron, gold amongst others) have been mined from prehistory into modern times [78,90,91], of which several traces are still preserved in the landscape. Next to mining, other traces of past human–environment interactions on the landscape can be observed such as old farms and agricultural terraces being abandoned in the mountains [92,93], intensive (greenhouse) agriculture in the coastal and alluvial plains [94–97] and tourism development in the coastal areas (littoralization) [98] with its large impacts on groundwater hydrology [96,98–100].

*2.2. Data Collection and Analysis*

For this study, 34 geosites were selected from southeastern Spain (see Figure 1b for their location; Table A1 in Appendix A for their description, Table A4 in Appendix A for their scenic beauty rating; Questionnaire S1 in the Supplementary Materials for their photos), which were part of an educational 'Physical Geography' field excursion, held from 18–24 May 2019. These geosites were selected based on their scientific value in order to present a variety of topics related to the geomorphology, geology, pedology, hydrology, and archaeology of the region, and with a strong focus on human–environment interactions in a Mediterranean environment at various timescales. Most geosites (30) are related to geomorphology. In addition, 27 of them have viewpoints while 16 geosites can be directly-linked to human–environment interactions. Specifically, the geosites include volcanic cones and columns, horst and graben structures, faults, travertine dams, sand dunes, tafoni, alluvial fans, landslides, gullies, badlands, mining, and archaeological settlements. Whilst scenic beauty did not form a major criterion when selecting the sites visited during the field trip, several geosites have been selected as they provide spectacular views to surrounding landscapes and are, hence, ideal sites for a physical geography field trip.

A total of 176 respondents (actual and potential tourists) participated in this survey (see Table A3 in Appendix A for their socio-demographic background). Of these, 104 were persons with no geoscience background and who did not visit the Spanish geosites (hereafter called NGB-NV); 43 were persons with a geoscience background and who did not visit the selected geosites (hereafter called GB-NV); and 29 were persons with a geoscience background and who all visited the selected geosites during the educational 'physical geography' field excursion, held in 2019 (hereafter called GB-V). The NGB-NV group consisted of persons whose educational background is unrelated to geosciences. On the other hand, the GB-NV and GB-V group comprised persons who studied geography and/or geology, and whose education level was bachelor's degree and higher. The GB-NV group was purposefully included to control for the effects of direct experience to the geosites and expert information on the scenic beauty rating between the NGB-NV and GB-V groups. The NGB-NV and GB-NV groups rated scenic beauty in an online survey based on representative photos of the geosites, while the GB-V group made the assessment after visiting

the geosites in 2019. The scenic beauty rating by these three groups was used to determine the contribution of geoscience knowledge for scenic beauty.

For the photo-based assessment, a total of 74 photos representing the 34 geosites were selected. These photos were presented in the order in which the GB-V group visited them in the field. Each geosite was represented with two to three photos to provide typical views of the geosites to the respondents. The first photo of each geosite showed an overview of the surrounding landscape, and the second (and third, if any) photo/s usually showed the geosite in more detail. From the photos used in the survey, 63 were photos taken during the field trip while the remaining 11 photos were taken from images in Google Earth and the internet due to the poor quality of some of the photos we had. While selecting the latter photos, care was taken to make them representative of what the GB-V group saw on-site and to illustrate in the best possible way the main geo-feature of the selected site.

The online survey had two sections. The first section consisted of photos and a five-point Likert scale (where 1 = not at all interesting, 2 = slightly interesting, 3 = moderately interesting, 4 = very interesting, and 5 = extremely interesting) for rating scenic beauty, while the second section comprised the socio-demographic background of the respondents (see Questionnaire S1, in the Supplementary Material).

The online survey was conducted from 18 February to 10 March 2021, and all authors of this study sent the online survey via email to people in their network. In some cases, people who were first contacted by the authors further distributed the survey to other people, and hence it was rather difficult to know the total respondents contacted. However, by counting those for which we had reliable data, it was estimated that the survey was distributed to over 550 people. A total of 154 completed surveys were received, of which, 7 were discarded because the respondents indicated that they had previously visited one or more of the geosites. Hence, the 147 completed responses (104 NGB-NV and 43 GB-NV) were used for further analysis of the scenic beauty of the geosites. The NGB-NV group comprised persons with educational background from ca. 25 disciplines such as archaeology, agriculture, biology, chemistry, engineering, economics, history, languages, management, medicine, psychology, sustainable development and tourism (see Figure A1 in Appendix A) while the GB-NV group consisted of 32 geographers and 11 geologists.

On the other hand, the GB-V group assessed the scenic beauty and scientific value of geosites during an educational field excursion to the Spanish study area. This group comprised a total of 32 students (from KU Leuven and Free University Brussels) who enrolled in the 3rd bachelor and 1st master in Geography. It also included two KU Leuven professors of physical geography (>20 years of experience in the region), who led the trip, and three field assistants with a master's degree and educational background related to geoscience. Hence, the scenic beauty and scientific value assessment questionnaire was distributed to the 37 participants of the field trip. A total of 29 persons completed the questionnaire, including the two professors and the three field assistants.

The GB-V group had learned about the geosites in southeastern Spain before and during the field excursion, which enabled them to evaluate the scientific value of each geosite. They collected and read research papers about the geosites and they also made short presentations in the classroom before the field excursion. In addition, they were also given on-site scientific information about the characteristics, genesis, importance for earth sciences as well as for human–environment interactions of each geosite by the two professors (Figure 2). Furthermore, shortly after visiting all sites, the GB-V group also evaluated the scenic beauty of the 34 geosites.

The GB-V group was asked to rate the scenic beauty and scientific value of the geosites using a five-point Likert scale (where 1 = not at all interesting, 2 = slightly interesting, 3 = moderately interesting, 4 = very interesting, and 5 = extremely interesting), which is a similar scale as that provided to the other two groups of respondents (see Questionnaire S2 and Questionnaire S3 in the Supplementary Materials). In addition, the GB-V group was also asked to list interesting features of each geosite related to its scenic beauty and scientific value. The socio-demographic profile of the respondents was also collected.

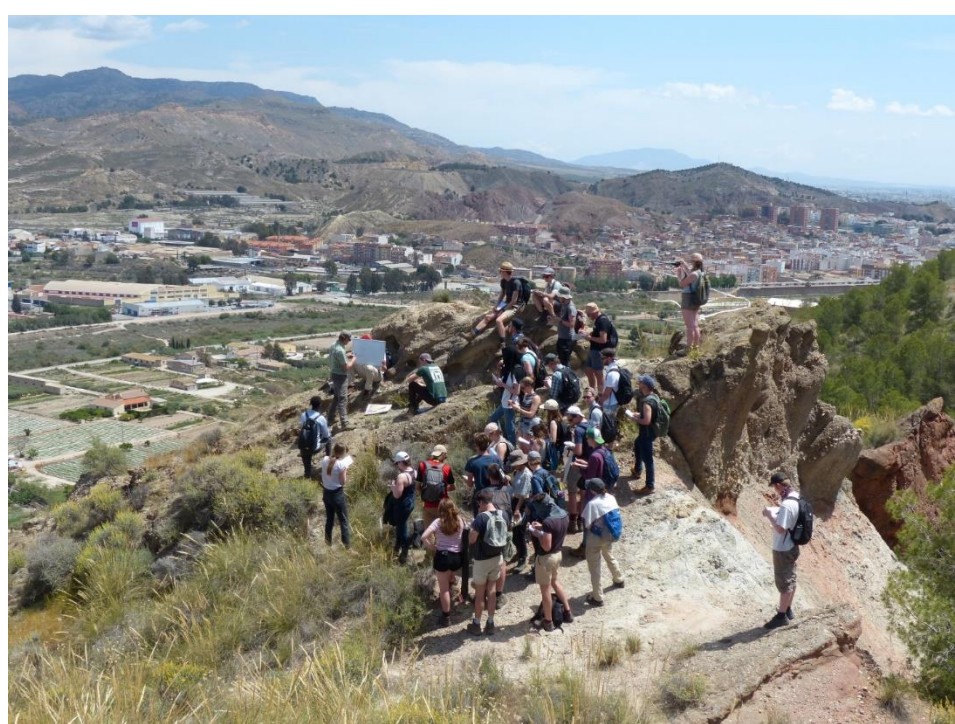

**Figure 2.** Scientific information being given to the GB-V group at Lorca (geosite 13; © J. Poesen, May 2019).

The GB-V group was briefed about the contents of the survey questionnaire in a classroom one day before departure to Spain. The questionnaire was distributed to them immediately before the start of the field excursion and collected at the end of the field excursion; this helped them to be familiar with the questions.

Correlation analysis and scatter plots and boxplots were made to test the relationship between the scenic beauty and scientific value of the geosites. In addition, to further investigate the relationship between scenic beauty and scientific value, a word cloud analysis was conducted. This revealed the most frequent keywords that the respondents reported in order to describe the interesting features of geosites reflecting their scientific value and scenic beauty. In the word cloud analysis, only keywords were selected, and co-occurring words were removed before the analysis. For example, if the respondent mentioned "view over sierra and sea", then the keyword 'view' was taken, and the words 'over', 'sierra' and 'sea' were dropped from the analysis. This helped to avoid unnecessary repetition of words.

In order to investigate the relationship between scenic beauty and geoscience knowledge, one-way ANOVA and post-hoc multiple comparisons were conducted for the mean scenic beauty ratings of the geosites among the three respondent groups. Additionally, to identify why some geosites received higher scenic beauty and scientific value ratings by the respondent groups, the 34 geosites were grouped into five clusters based on the presence of particular features of interest to tourists at each geosite. The features used as criteria were local geo-features (such as volcanic cones and columns, horst and graben structures, faults, travertine dams, sand dunes, tafoni, alluvial fans, landslides, gullies, etc.), human–environment interaction (such as archaeological sites, agricultural fields-both currently in use and abandoned ones, dams and reservoirs) and viewpoints. The resulting clusters of geosites, in order of their numbers of geosites they have, are (1) HE = human–environment interaction features (2 geosites); (2) HE-LG-VP = human–environment interaction feature, local geo-feature and viewpoint (4 geosites); (3) LG = local geo-feature (5 geosites); (4) HE-VP = human–environment interaction feature and viewpoint (10 geosites); and

(5) LG-VP = local geo-feature and viewpoint (13 geosites) (see Table A2 in Appendix A for the list of clustered geosites).

## 3. Results

### 3.1. Scenic Beauty Rating and Socio-Demographic Background of Respondents

Table 1 shows the mean scenic beauty ratings of the 34 geosites as a function of socio-demographic factors. It can be seen that, on average, women (mean = 3.35; sd = 0.51) rated scenic beauty relatively higher than men (mean = 3.25; sd = 0.52). Young people with age 18–29 (mean = 3.31; sd = 0.57) rated scenic beauty higher than other age groups. In terms of education, those with bachelor's degree (mean = 3.34; sd = 0.53) rated scenic beauty higher than those with master's and PhD degrees. Besides, those who live in Belgium (mean = 3.46; sd = 0.6) rated scenic beauty higher than those from other countries. Furthermore, those who did not travel outside their continent (mean = 3.42; sd = 0.47) rated the scenic beauty of the geosites higher than respondents who visited one or more other continents. It was also found that respondents with geoscience background and who visited the geosites (mean = 3.40; sd = 0.34) rated scenic beauty higher than those with no geoscience background and who did not visit the geosites. As to the respondents' general primary preferred types of attractions, those who prefer attractions related to biodiversity and geology-landscape (mean = 3.6; sd = 0.56) rated scenic beauty higher than other groups.

**Table 1.** Mean scenic beauty ratings (scale range 1–5; standard deviations in parentheses) of the 34 geosites as a function of socio-demographic factors of the respondents (*n* is the number of persons). For information on the socio-demographic background of the respondents, see Table A3 in Appendix A.

| Socio-Demographic Factor | | Mean | F Value | *p* Value |
|---|---|---|---|---|
| Gender | Female (*n* = 71) | 3.35 (0.51) | 1.73 | 0.19 |
| | Male (*n* = 99) | 3.25 (0.52) | | |
| Age [a] | 18–29 (*n* = 71) | 3.31 (0.57) | 0.83 | 0.48 |
| | 30–49 (*n* = 35) | 3.25 (0.46) | | |
| | 50 and above (*n* = 41) | 3.22 (0.55) | | |
| Education level | Bachelor's degree (*n* = 54) | 3.34 (0.53) | 0.31 | 0.74 |
| | Master's degree (*n* = 98) | 3.28 (0.51) | | |
| | PhD degree (*n* = 17) | 3.26 (0.49) | | |
| Country of residence [b] | Belgium (*n* = 125) | 3.46 (0.6) | 1.39 | 0.25 |
| | Other (*n* = 51) | 3.33 (0.52) | | |
| Continents visited | No travel outside my continent (*n* = 48) | 3.42 (0.47) | 1.76 | 0.16 |
| | Visited one other continent (*n* = 54) | 3.24 (0.53) | | |
| | Visited 2–3 other continents (*n* = 44) | 3.29 (0.54) | | |
| | Visited 4–5 other continents (*n* = 29) | 3.16 (0.46) | | |
| Respondent group [c] | NGB-NV (*n* = 104) | 3.22 (0.53) | 2.50 | 0.08 [d] |
| | GB-NV (*n* = 43) | 3.39 (0.53) | | |
| | GB-V (*n* = 29) | 3.40 (0.34) | | |
| Primarily preferred attraction type/s | Biodiversity (*n* = 13) | 3.35 (0.48) | | |
| | Biodiversity and culture-history (*n* = 11) | 3.1 (0.49) | | |
| | Biodiversity, culture-history, and geology-landscape (*n* = 31) | 3.44 (0.58) | | |
| | Biodiversity and geology-landscape (*n* = 3) | 3.6 (0.56) | | |
| | Culture-history (*n* = 37) | 3.09 (0.56) | | |
| | Culture-history and geology-landscape (*n* = 27) | 3.37 (0.43) | | |
| | Geology-landscape (*n* = 51) | 3.31 (0.45) | 2.05 | 0.06 [d] |

[a] Does not include data of those who visited the geosites. [b] Includes temporary (e.g., students) and permanent residence. [c] Respondent group refers to respondents grouped following their geoscience background and field visits: NGB-NV = no geoscience background and no visit to geosites; GB-NV = geoscience background and no visit to geosites; GB-V = geoscience background and visit to geosites. [d] Significant at *p* < 0.1; scenic beauty rated on 1–5 scale.

However, one-way ANOVA results showed that gender, age, country of residence and number of continents visited did not significantly influence mean scenic beauty rating while the respondent group and primarily preferred attraction were found significant at $p < 0.1$ (Table 1, Figure 3). Post-hoc pairwise (multiple) comparisons showed that none of the socio-demographic factors, including the respondent group and primarily preferred attraction, revealed a significant difference (at $p < 0.1$) in the scenic beauty ratings of the 34 geosites.

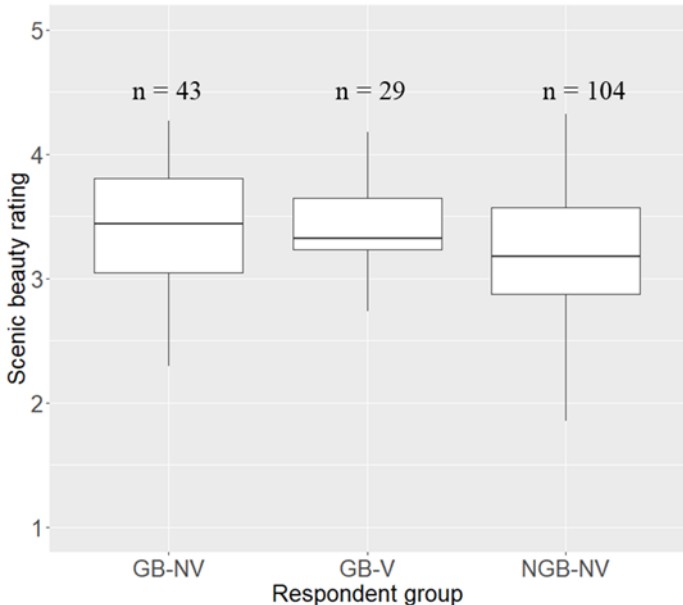

**Figure 3.** Boxplot of mean scenic beauty ratings of the 34 geosites by respondents, grouped following their geoscience background and field visits: NGB-NV = no geoscience background and no visit to geosites; GB-NV = geoscience background and no visit to geosites; GB-V = geoscience background and visit to geosites; $n$ = the number of respondents in each group; scenic beauty was rated on a 1–5 scale.

To further investigate the effect of socio-demographic factors (excluding age as there was no data for the GB-V group in the "respondent group" factor) on scenic beauty rating, two-way ANOVA was conducted. Only the interaction of education level and country of residence were found significant (F value = 3.56, $p < 0.05$). However, further pairwise multiple comparisons indicated that none of the interaction effects were significant at $p < 0.1$.

### 3.2. Relationship between Scenic Beauty and Scientific Value of Geosites

3.2.1. Correlation Analysis Results

Geosites with a higher scientific value were rated higher for their scenic beauty, and those with lower scientific value were also rated lower by all the respondent groups (Figures 4a–c and 5a–c). In addition, the mean scenic beauty rating by all the respondent groups is positively related to the scientific value of geosites (Figure 5d).

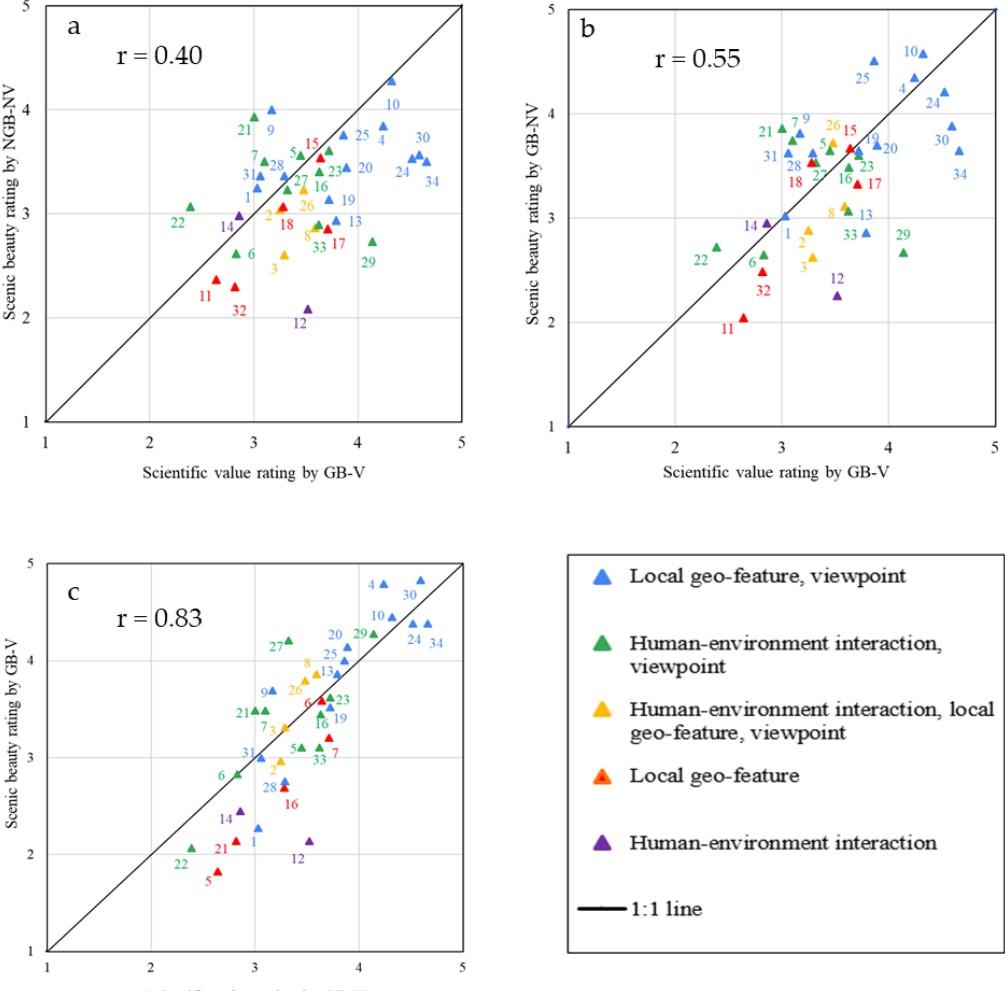

**Figure 4.** (**a–c**) Relationship between scenic beauty ratings of the 34 geosites (rated by the three groups of respondents, grouped following their geoscience background and field visits: NGB-NV = no geoscience background and no visit to geosites; GB-NV = geoscience background and no visit to geosites; GB-V = geoscience background and visit to geosites) and scientific value (rated by the GB-V group). Both scenic beauty and scientific value were rated on a 1–5 scale. r = Pearson's correlation coefficient between scenic beauty (as rated by each respondent group) and scientific value (rated by the GB-V group). See Figure 1b for the location of geosites (indicated by their number in the figure); Table A1 in Appendix A for their description, Table A4 in Appendix A for their mean scenic beauty ratings; Table A5 in Appendix A for their scientific value ratings; Questionnaire S1 in the Supplementary Materials for their photos.

Correlation analysis also indicated that there is a significant (at least at $p < 0.05$) positive relationship between the scenic beauty and scientific value of the geosites (Figures 4a–c and 5a–c). However, the strength of the relationship depends on the type of respondent group who rated the scenic beauty. A weak relationship (Figure 4a) was found between scenic beauty rating by the NGB-NV group and scientific value (rated by the GB-V group) while a moderate relationship (Figure 4b) was found between scenic beauty as rated by the GB-NV group and scientific value (as rated by the GB-V group). The relationship between the scenic beauty and scientific value of the geosites, both rated by the GB-V group, was strong (Figure 4c). Furthermore, a positive relationship (r = 0.70) was found between the mean scenic beauty ratings of the geosites by all the three respondent groups and the scientific value ratings of the geosites.

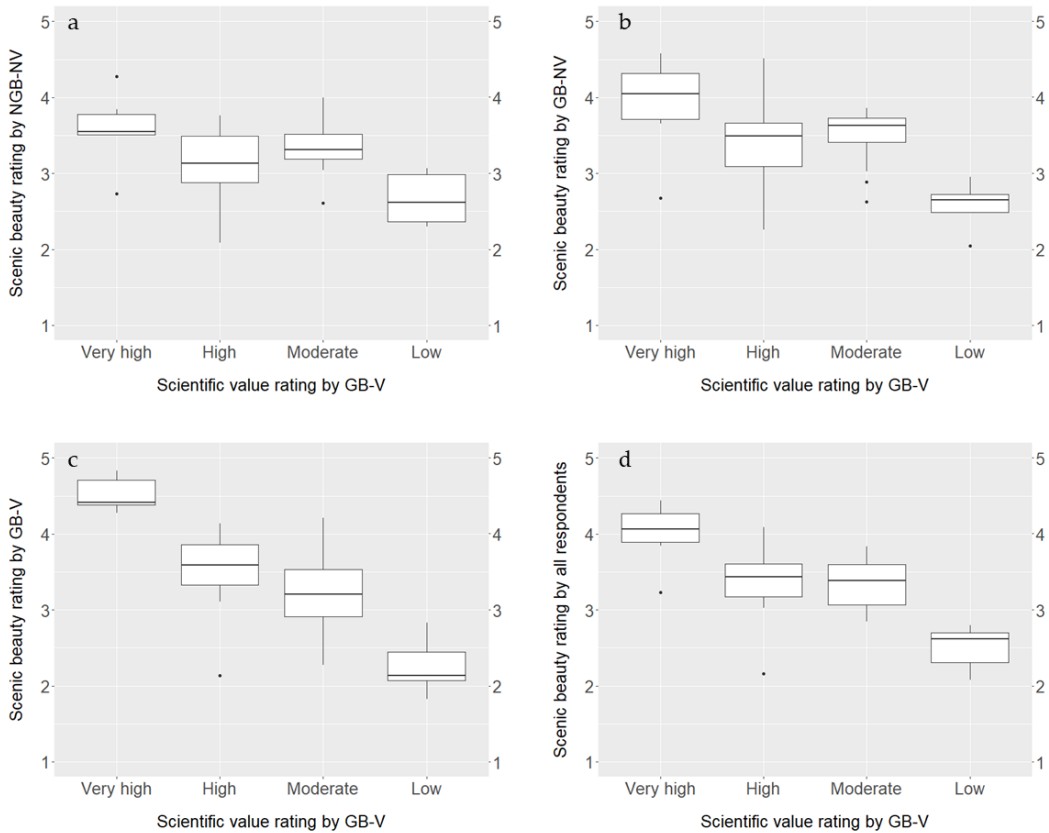

**Figure 5.** (**a**–**d**) Boxplots of scenic beauty rating by respondents, grouped following their geoscience background and field visits (NGB-NV = no geoscience background no visit to geosites, 104 persons; GB-NV = geoscience background and no visit to geosites, 43 persons; GB-V = geoscience background and visit to geosites, 29 persons), for different scientific value ratings. Scientific value was rated by the GB-V group (29 persons) on a 1–5 scale. This scientific value rating of geosites was then grouped based on mean scientific value rating: Very high = 4.0 and above (6 geosites); High = 3.5–3.9 (11 geosites); Moderate = 3.0–3.5 (12 geosites); Low = below 3.0 (5 geosites). The scientific value rating was done by the GB-V group, 29 persons (see Table A5 in Appendix A). "Scenic beauty rating by all respondents" is the mean scenic beauty rating of the three respondent groups.

### 3.2.2. Interesting Features Explaining Scenic Beauty and Scientific Value Ratings of the Geosites

The GB-V group was asked to list the most interesting geo-features that make up the scenic beauty as well as the scientific value of the geosites (Figure 6). The top five most frequent reported words describing interesting scenic features (Figure 6a) were view = 146, landscape = 38, gully = 34, sea = 32, and travertine = 31.

The top five most frequent words which the respondents mentioned as interesting features of geosites contributing to their scientific value (Figure 6b) were gully = 33, travertine = 31, terrace = 30, dam = 29, and badland = 21.

Among the most frequent words in the word cloud analysis, 22 are common to both scenic beauty and scientific value (which account for 45.8% of the words in Figure 6a and 57.9% in Figure 6b). These include, in alphabetical order, archaeology, atoll, badland, biodiversity, caldera, castle, dam, dune, flood, geology, gully, history, landslide, mining, rambla, rock, tafoni, terrace, travertine, valley, viewpoint and wind. These common words indicate that the GB-V respondent group appreciates scenic beauty and scientific value on many similar features, supporting our hypothesis that these two geosite values are interrelated.

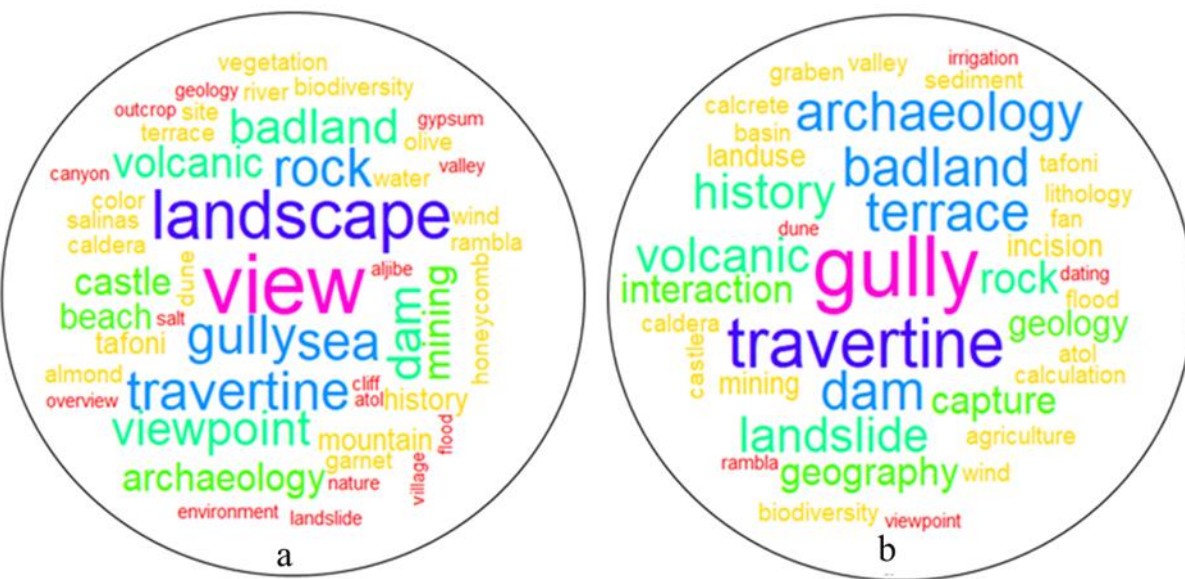

**Figure 6.** The most frequently reported features of geosites explaining their scenic beauty and scientific value, as mentioned by the GB-V group (29 persons). (**a**) scenic beauty (total words = 48, minimum and maximum word frequency = 5 and 146, respectively); (**b**) scientific value (total words = 38, minimum and maximum word frequency = 5 and 33, respectively). Note that the word size corresponds to the frequency of that word in its category, i.e., in scenic beauty or scientific value.

### 3.3. Contribution of Geoscience Knowledge to Perceived Scenic Beauty

As shown in Figure 7, there is a relative agreement among the three respondent groups in the scenic beauty rating. More specifically, from the three pairwise comparisons, there is a better agreement in scenic beauty ratings of the geosites by the NGB-NV and GB-NV groups as their ratings are close to the 1:1 line.

However, in absolute terms, the groups with geoscience background (GB-NV and GB-V) gave higher scenic beauty rating to more geosites (i.e., to 24 and 20 geosites, respectively) as compared to the NGB-NV group (Figure 7). On the other hand, the GB-NV group gave a higher scenic beauty rating to 19 geosites and vice versa for the remaining 15 geosites (Figure 7b).

The data also revealed that geosite clusters where viewpoints are present were among those that were rated higher for their scenic beauty as well as for their scientific value (Figure 8). More specifically, all respondent groups rated scenic beauty higher for geosites that have both a local geo-feature and offer a viewpoint (Figure 8a–c). In addition, the scientific value was also rated higher by the GB-V group for this cluster of geosites (Figure 8d).

One-way ANOVA was used to compare the mean scenic beauty ratings of the 34 individual geosites by the three respondent groups (Figure 9). Significant differences were found for 22 individual geosites at different significance levels: $p < 0.05$, $p < 0.01$ and $p < 0.001$ (see Table A4 in Appendix A).

Post-hoc pairwise multiple comparisons showed significant mean scenic beauty rating differences between the three respondent groups (Table 2). A significant difference in scenic beauty rating was found between GB-NV and NGB-NV groups for 7 out of 34 geosites. For all these geosites, the GB-NV group rated scenic beauty higher than the NGB-NV group. In addition, a significant difference in scenic beauty rating was found between NGB-NV and GB-V for 17 out of 34 geosites. For 11 of these geosites, the GB-V group rated scenic beauty higher than the NGB-NV group, and vice versa for the remaining six geosites. Furthermore, a significant difference in mean scenic beauty rating was found between GB-NV and GB-V groups for 14 out of 34 geosites. For eight of these geosites, the GB-V group rated scenic beauty higher than the GB-NV group and vice versa for the rest six geosites.

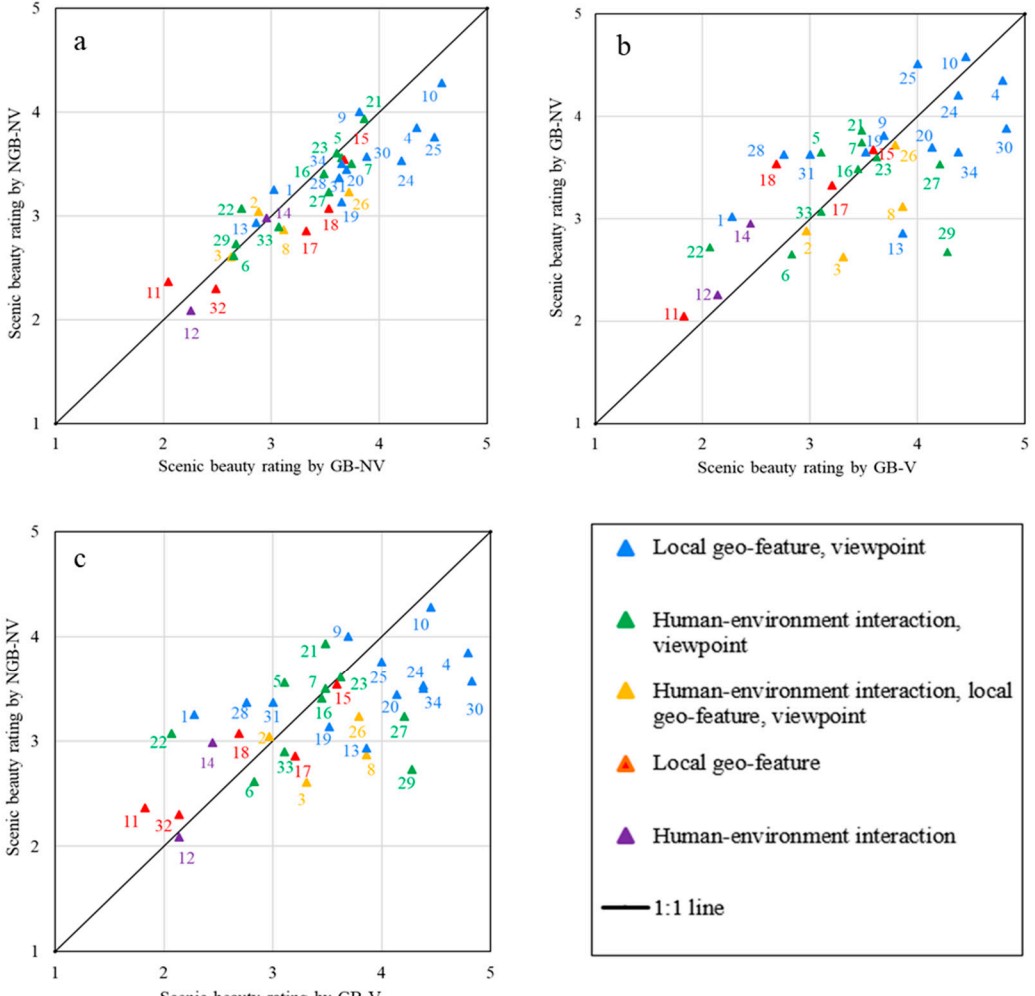

**Figure 7.** (**a**–**c**) Comparison of mean scenic beauty ratings of the 34 geosites by the three respondent groups, grouped following their geoscience background and field visits: NGB-NV = no geoscience background and no visit to geosites; GB-NV = geoscience background and no visit to geosites; GB-V = geoscience background and visit to geosites. Scenic beauty was rated on a 1–5 scale. See Figure 1b for the location of geosites (indicated by their number in the figures); Table A1 in Appendix A for their description, Table A4 in Appendix A for their mean scenic beauty rating; Questionnaire S1 in the Supplementary Materials for their photos.

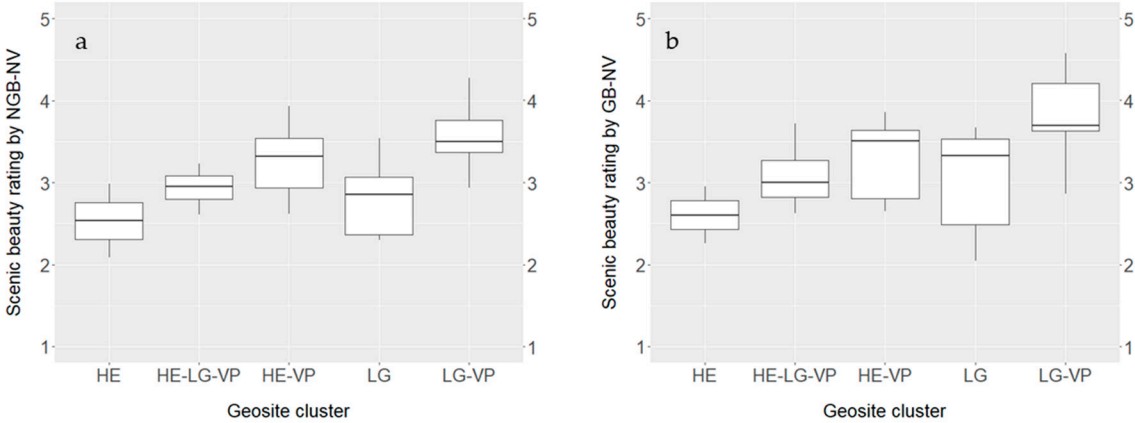

**Figure 8.** *Cont.*

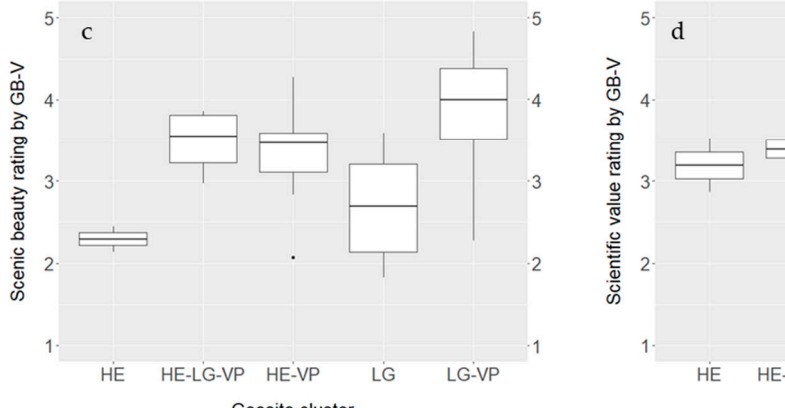
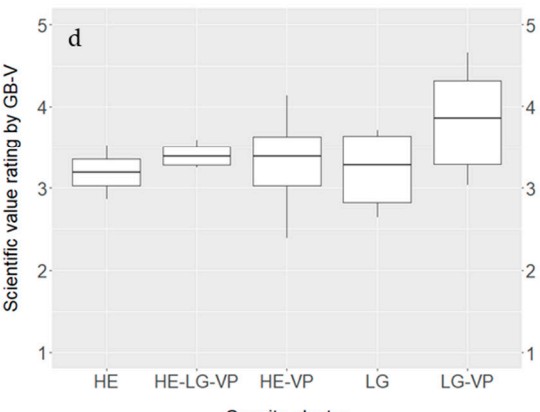

**Figure 8.** (**a**–**d**) Boxplots of scenic beauty rating by respondents, grouped following their geoscience background and field visits (NGB-NV = no geoscience background no visit to geosites, 104 persons; GB-NV = geoscience background and no visit to geosites, 43 persons; GB-V = geoscience background and visit to geosites (29 persons), for different geosite clusters: LG-VP = local geo-feature and viewpoint, 13 geosites; HE-VP = human–environment interaction feature and viewpoint, 10 geosites; HE-LG-VP = human–environment interaction feature, local geo-feature and viewpoint, 4 geosites; LG = local geo-feature, 5 geosites; HE = human–environment interaction feature, 2 geosites). See Table A2 in Appendix A for the list of clustered geosites. The scientific value rating was done by the GB-V group (29 persons). Both scenic beauty and scientific value were rated on a 1–5 scale (see Tables A4 and A5 in Appendix A for their mean ratings).

**Table 2.** Pairwise multiple comparisons of mean scenic beauty ratings (standard deviations in parentheses) of the 22 individual geosites (where a significant difference in mean scenic beauty rating was found) by the three groups, grouped following their geoscience background and field visits: NGB-NV = no geoscience background and no visit to geosites; GB-NV = geoscience background and no visit to geosites; GB-V = geoscience background and visit to geosites. Scenic beauty rated on a 1–5 scale.

| Geosite Number | Geosite Name | Mean of NGB-NV (a) | Mean of GB-NV (b) | Mean of GB-V (c) | Mean Difference | | |
|---|---|---|---|---|---|---|---|
| | | | | | (b–a) | (c–a) | (c–b) |
| 1 | Boca Andarax | 3.25 (0.93) | 3.02 (1.01) | 2.28 (0.92) | −0.23 | −0.97 *** | −0.74 ** |
| 3 | Las Salinas | 2.61 (0.84) | 2.63 (1.05) | 3.31 (1) | 0.02 | 0.70 *** | 0.68 ** |
| 4 | Punta Baja | 3.85 (0.89) | 4.35 (0.61) | 4.79 (0.41) | 0.50 ** | 0.95 *** | 0.44 ** |
| 5 | Cerro Pistolas | 3.56 (0.9) | 3.65 (0.9) | 3.10 (1.11) | 0.09 | −0.45 * | −0.55 * |
| 8 | Rodalquilar Mine | 2.87 (1.01) | 3.12 (1.1) | 3.86 (0.79) | 0.25 | 0.99 *** | 0.75 ** |
| 11 | El Puntal | 2.37 (0.89) | 2.05 (1.07) | 1.83 (1.07) | −0.32 | −0.54 ** | −0.22 |
| 13 | Lorca Castle | 2.93 (0.98) | 2.86 (1.04) | 3.86 (0.83) | −0.07 | 0.93 *** | 1.00 *** |
| 14 | Puentes Dam | 2.98 (0.9) | 2.95 (1.17) | 2.45 (1.02) | −0.03 | −0.53 * | 0.50 |
| 17 | Rio Alias | 2.86 (1) | 3.33 (0.94) | 3.21 (0.9) | 0.47 * | 0.35 | −0.11 |
| 18 | Rambla de los Feos | 3.07 (0.95) | 3.53 (0.93) | 2.69 (0.85) | 0.46 * | −0.38 | −0.85 *** |
| 19 | Los Perales | 3.13 (0.98) | 3.65 (1.15) | 3.52 (1.02) | 0.52 * | 0.38 | −0.13 |
| 20 | Los Molinos | 3.44 (0.98) | 3.7 (1.08) | 4.14 (0.92) | 0.26 | 0.70 ** | 0.44 |
| 22 | Los Yesos | 3.07 (1.05) | 2.72 (1.05) | 2.07 (0.96) | −0.35 | −0.99 *** | −0.65 * |
| 24 | Bar Alfaro | 3.53 (0.99) | 4.21 (0.94) | 4.38 (0.86) | 0.68 *** | 0.85 *** | 0.17 |
| 25 | Mini Hollywood | 3.76 (1.03) | 4.51 (0.63) | 4 (0.89) | 0.75 *** | 0.24 | −0.51 |
| 26 | Rambla Honda | 3.23 (1.04) | 3.72 (0.91) | 3.79 (1.01) | 0.49 * | 0.56 * | 0.02 |
| 27 | La Calahorra | 3.23 (1.05) | 3.53 (0.98) | 4.21 (0.82) | 0.30 | 0.98 *** | 0.67 * |
| 28 | Esfiliana | 3.37 (1.01) | 3.63 (1.05) | 2.76 (0.95) | 0.26 | −0.61 * | −0.87 ** |
| 29 | Gorafe | 2.73 (0.93) | 2.67 (0.99) | 4.28 (0.7) | −0.06 | 1.55 *** | 1.60 *** |
| 30 | Alicun de las Torres | 3.57 (0.97) | 3.88 (0.79) | 4.83 (0.54) | 0.32 | 1.26 *** | 0.94 *** |
| 31 | Belerda | 3.37 (0.88) | 3.63 (1.16) | 3 (1) | 0.26 | −0.37 | −0.63 * |
| 34 | El Hoyazo | 3.5 (1.01) | 3.65 (0.92) | 4.38 (0.56) | 0.15 | 0.88 *** | 0.73 ** |

* $p < 0.05$; ** $p < 0.01$; ** $p < 0.001$.

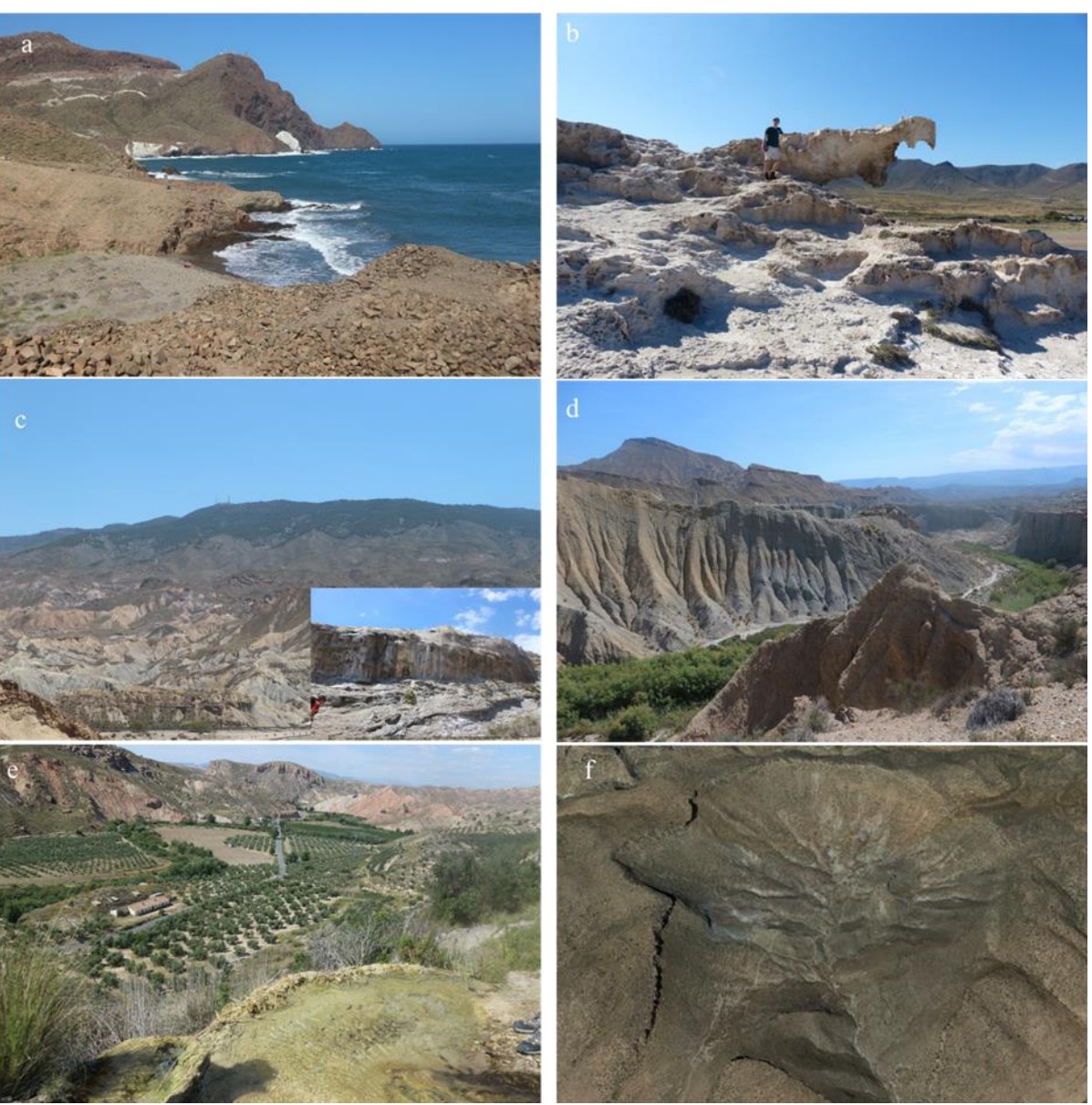

**Figure 9.** Illustration of the top-ranked geosites based on mean scenic beauty ratings by the three respondent groups. (**a**) Punta Baja (geosite 4) with andesitic columns (© G. Tessema, May 2019); (**b**) Los Escullos (geosite 10) with fossil dunes (© J. Poesen, May 2019); (**c**) Bar Alfaro (geosite 24) view on badlands near rambla de Tabernas with exposures of flysch and marls, inset photo—travertine features (© G. Tessema, May 2019); (**d**) Mini Hollywood (geosite 25) with view on badlands near rambla de Tabernas (© G. Tessema, May 2019); (**e**) Alicun de las Torres (geosite 30) with flowing water on travertine (© G. Tessema, 2019. (**f**) El Hoyazo (geosite 34)—eroded volcanic cone and coral reef (atoll) (© Google Earth Images, May 2020). These geosites are also among those in the top rank in terms of scientific value rating. See Figure 1b for their location; Table A1 in Appendix A for their description, Table A4 in Appendix A for their scenic beauty ratings; Questionnaire S1 in the Supplementary Materials for their photos.

## 4. Discussion and Conclusions

The objective of this research was to better understand the relationships between scenic beauty of geosites, their scientific value as well as geoscience knowledge of actual and potential tourists. The scenic beauty of 34 geosites in southeastern Spain was evaluated by 176 respondents, which were divided into three groups based on their geoscience background and visits to the geosites.

Socio-demographic factors such as gender, age, education level, country of residence and number of continents visited did not significantly affect the scenic beauty ratings of the geosites by the respondents. Previous studies on the assessment of scenic beauty of landscapes [43,55,62,63] also found that age, gender and education did not significantly affect the scenic beauty ratings of landscapes. In addition, Frank et al. [42] also found that age, gender and personal qualification (i.e., layman, stakeholder and expert) did not significantly influence landscape scenic beauty assessment results. On the other hand, Skřivanová et al. [61] found that there is a significant difference in the scenic beauty rating of landscapes between women and men, the former generally rating it higher. Svobodova et al. [50] studied the visual preferences for physical attributes of mining and post-mining landscapes with 1050 persons in a web-based survey in the Czech Republic and found that visual preferences for landscapes significantly varied based on gender, education level and professional field or study focus. Women rated the scenic beauty of landscapes significantly higher than men. Those with education lower than university-level rated scenic beauty of landscapes significantly higher than those with a university degree. Moreover, respondents whose profession is not related to landscape management rated scenic beauty of landscapes higher than respondents whose education is related to landscape management (e.g., ecology, nature conservation, architecture). López-Martínez [55] pointed out that different findings among studies about the effect of socio-demographic factors on scenic beauty assessments of landscapes might be attributed to differences in study areas, and thereby landscapes being evaluated.

The correlation analysis in this study revealed that there is a positive relationship between scenic beauty and scientific value of geosites, and this relationship improved with an increase in geoscience knowledge of the tourists and with a field visit of these sites. The positive correlation between scenic beauty and scientific value was stronger for the GB-V group than the GB-NV group, and a possible source of difference would be that the former rated the scientific value, received scientific information on-site and visited the geosites. This positive correlation was stronger for both the GB-V and GB-NV groups in comparison with the NGB-NV group, which can be attributed to the geoscience education of both GB groups.

The word cloud analysis of keywords provided by the respondents to describe the most interesting features at each geosite for their scenic beauty and scientific value supports the finding that scenic beauty and scientific value are interrelated. It should, however, be noted in the word cloud analysis that a larger frequency of keywords does not necessarily imply that more respondents reported a particular word (feature); it might well be that few respondents repeatedly mentioned such a word. Moreover, a respondent might have been interested in more than one type of interesting feature for a given geosite, and hence the frequency of the words might be larger.

There is a general consensus among the respondents in their scenic beauty rating in that for most cases, geosites rated higher by one group were also rated higher by the other (see Figure 7). Kalivoda et al. [38] studied scenic beauty rating between experts and non-experts and found that the higher the scenic beauty ranking, the better the consensus between the two groups. They argued that such a consensus plays an important role for the legal protection of geosites.

Overall, the GB-NV and GB-V groups rated the scenic beauty of some geosites significantly higher than the NGB-NV group (Table 2), and this could be because their geoscience knowledge helped them to better appreciate geosites than the NGB-NV group. Reynard and Giusti [101] (p. 152) support this arguing that "perhaps the [scenic] beauty resides . . . less in the outburst of emotions than in the elements of understanding". In addition, the field visit to the geosites by the GB-V group could also have helped them to rate scenic beauty compared to the NGB-NV group.

The GB-V group rated some geosites significantly higher than the GB-NV group, which could be attributed to the expert information the former received while visiting the geosites. Obviously, the GB-V group has better geoscience knowledge about the geosites

in this study than the GB-NV group due to the desk research the former had made about the geosites and the expert information it had received on-site. In addition, the field visit could also have had an impact on the scenic beauty rating, allowing the GB-V group to rate scenic beauty higher than the GB-NV group.

In addition to geoscience knowledge, the field visit and the relatively good weather conditions (dry and sunny) during the visit could also explain why the GB-V group rated some geosites significantly higher than the NGB-NV and GB-NV groups. The field visit allowed to see more detailed features of the geosites than the photos. Hull and Stewart [102] argued that photographs may not always capture all the landscape differences, allowing those who visit the sites to rate scenic beauty higher than the group who does not visit.

A good example where the on-site expert information and the field visit might have caused a significantly higher scenic beauty rating of the geosites by the GB-V group than the other two groups are the Gorafe and Alicun de las Torres geosites (geosite numbers 29 and 30, Table 2). Among the interesting features of Gorafe are the view over the Rio Gor valley cut into the Guadix-Baza sedimentary basin and the nearby Bronze-age megalithic park. The latter could not be clearly understood unless one travels to this geosite and also receives expert information about its archaeological history and significance. In addition, during the field excursion, the unique travertine features of Alicun de las Torres were shown and explained to the GB-V group and they were also able to appreciate the flowing spring water and landscape on-site.

On the other hand, the NGB-NV and GB-NV groups rated some geosites significantly higher than the GB-V group (for example, geosite numbers 1, 5, 22 and 28). These were among the lowest-rated geosites in terms of their scientific value by the GB-V group. As scientific value has a relationship with scenic beauty, it might also have contributed to a lower appreciation of these geosites. In other words, the on-site scientific interpretations provided to the GB-V group might have influenced them not to rate these geosites higher for their scenic beauty. For example, at Cerro Pistolas and El Puntal (geosites 5 and 11, respectively), the GB-V group received scientific information about the impacts of the greenhouses on the landscape and the related landscape degradation processes (land levelling, groundwater extraction and pollution by pesticides and degraded plastics), and this might have negatively affected their scenic beauty rating. There was also litter dumped at El Puntal which could have influenced the scenic beauty rating by the GB-V group. Stecker [67] argues that though scientific knowledge could enhance scenic beauty appreciation, it could also work the other way round, i.e., "knowledge prevents appreciation from being malfounded, from appreciating a part of nature for properties it does not have" (p. 400).

The findings in this study that scenic beauty and scientific value of geosites are interrelated and that geoscience knowledge contributes to higher perceived scenic beauty have important implications for geoconservation, geoheritage management and geotourism development. Erikstad [103] indicated that the necessity of geoconservation is not well developed and accepted in many countries around the world. The relationship between the two values could provide a strong support for the protection of the geosites. In addition, more geoscience education and geo-interpretation to people could help them to appreciate the scenic beauty of geosites. Beck et al. [104] indicated that one of the main objectives of interpretation is creating appreciation and deeper understanding of nature. It has also been argued that applying the concept of geotourism and geosites to particular landforms is the best way to transfer geoscience knowledge to society [17]. Thus, the geo-interpretations offered could help enhance visitor experience, thereby contributing to sustainable geotourism development. According to Newsome and Dowling [105] (p. 6), "visitors will always rate their experiences higher if they have also learned something about the landscape and geology they are visiting". People who appreciate the scenic beauty of geosites could in turn play their part for the conservation and management of these geosites. Education about the geosites also raises awareness for their protection [106].

The fact that geosites combining certain features are more interesting to all types of respondents, irrespective of their geoscience background, is important for geotourism development. The presence of a viewpoint was an important factor in the scenic beauty rating. Mikhailenko and Ruban [107] also indicated that the value of viewpoints (also called viewpoint geosites) in the western Caucasus (Russia), is strongly linked to their aesthetic properties. This study demonstrated that geosites combining a local geo-feature and a viewpoint were the most preferred by all groups of respondents. A study conducted in the Lake Tana region in Ethiopia also found that geosites combining a local geo-feature and a viewpoint were rated highest for their scenic beauty [16]. In addition, a survey of 582 visitors in the Albacete mountains (Spain), indicated that among the components that shape the character of the landscape (such as relief, water, vegetation, rural habitat, the combination of human and natural environments, climate, rural landscape, environmental quality and wildlife), the most important landscape component (as indicated by the rankings of these components by the respondents) was relief which includes mountains, gorges and valleys [108]. Such landscape components combine a local geo-feature and a viewpoint, and hence also support our finding. Geosites that combine a local geo-feature and a viewpoint can cater to the needs of different types of geotourists: from those that are purely interested in the geo-feature to those that just want to appreciate the scenic beauty of geosites.

Although there can be factors which might influence the perception of observers in a photo-based scenic beauty assessment such as the height of the horizon in the photograph and the shape of the photograph (e.g., square vs. wide angle), we believe that this had a minor impact on scenic beauty ratings of the geosites in our study. This is because the representative photographs of the geosites were carefully selected from many photos the first and second authors of this paper took during the field excursion in 2019. Where we believed that our photos were not representative, we selected some photos from previous excursions to the study area or from the internet. In addition, we showed two to three representative photos of each geosite to the respondents, in order to provide them with a typical view of the geosite. Moreover, the photo-based survey was conducted online using "Google Forms" and one photo was displayed per page and hence no downsizing of photos was made. In addition, we based our answers to hypothesis two mainly on the comparison of scenic beauty ratings between the NGB-NV and GB-NV groups, who rated scenic beauty based on photos of the geosites.

We acknowledge the limitation of our research in that the geosites selected were mainly related to geomorphology and human–environment interactions, and therefore lack more diversity. The number of participants ($n = 176$) in the survey was also relatively small and less diverse in their socio-demographic background. Moreover, the NGB-NV group rated scenic beauty based on photographs only. Future research could investigate scenic beauty rating with the following settings: (1) larger sample size of respondents and more diverse socio-demographic profiles; (2) more diverse geosites; (3) respondents with geoscience and no-geoscience background both groups visiting geosites in real life and one group offered scientific information and the other not, and repeating this for multiple groups; (4) scientific value of geosites rated by persons other than those who rate the scenic beauty. Although our study reveals some important relations, future research taking these recommendations into account will allow the drawing of more general conclusions about the relationship between scenic beauty and scientific value, as well as the contribution of geoscience knowledge to perceived scenic beauty rating.

**Supplementary Materials:** The following are available online at https://www.mdpi.com/article/10.3390/land10050460/s1, Questionnaire S1: Questionnaire for rating the scenic beauty of the 34 geosites by the NGB-NV and GB-NV groups, based on representative photographs of the geosites, Questionnaire S2: Questionnaire for rating the scenic beauty of the 34 geosites by the GB-V group, based on field visit. Questionnaire S3: Questionnaire for rating the scientific value of the 34 geosites by the GB-V group, based on field visit.

**Author Contributions:** Conceptualization, G.A.T. and J.P.; methodology, A.V.R., G.A.T., G.V. and J.P.; formal analysis, G.A.T. and J.P.; field excursion, G.A.T., J.P. and G.V.; data collection, A.V.R., G.A.T., G.V., J.P. and J.v.d.B.; writing—original draft preparation, G.A.T.; writing—review and editing, A.V.R., G.V., J.P. and J.v.d.B.; funding acquisition, G.V., J.P. and J.v.d.B.; supervision, A.V.R., J.P. and J.v.d.B. All authors have read and agreed to the published version of the manuscript.

**Funding:** The field excursion was sponsored by KU Leuven and VLIR-UOS project.

**Institutional Review Board Statement:** Not applicable.

**Informed Consent Statement:** Not applicable.

**Data Availability Statement:** The data presented in this study are available on request from the corresponding author. The questionnaire and photos of geosites used in this study are available in the supplementary material.

**Acknowledgments:** The authors thank the Institutional University Cooperation-Bahir Dar University (IUC-BDU) VLIR-UOS project for providing a PhD. research fellowship to the first author. We express our gratitude to Marie Rose Poesen and Ashebir Sewale who helped in distributing the online questionnaire to potential respondents. We also thank the 176 respondents who provided valuable input via the questionnaire. The authors are also grateful to the two anonymous reviewers for their constructive comments.

**Conflicts of Interest:** The authors declare no conflict of interest.

## Appendix A

**Table A1.** Description of the geosites in southeastern Spain. For their location, see Figure 1b; for their scenic beauty rating, see Table A3 in Appendix A; for their photos, see Questionnaire S1 in the Supplementary Materials.

| Geosite Number [a] | Name of the Geosite, and/or Town | Major Features |
|---|---|---|
| 1 | Boca Andarax, Almeria | Delta of the Andarax river, coastal erosion and Holocene environmental change. Viewpoint on sierras and pediments. |
| 2 | Torre Garcia, Retamar | Marine terraces, Palomares fault, vegetated sand dunes, Rambla de las Amoladeras, archaeological site where garum (fermented fish sauce) was produced in Roman times. |
| 3 | Las Salinas, La Fabriquilla | Lagoon with salt production basins (Salinas), alluvial fans at the foot of Sierra de Gata |
| 4 | Punta Baja, Cabo de Gata | Volcanic plug and quarry with exposure of andesitic columns. |
| 5 | Cerro Pistolas, El Nazareno | Viewpoint over Sierra de Gata and Nijar basin, traditional (Cortijo) and modern (greenhouses) land use (littoralisation) |
| 6 | Albaricoques | Sierra de Gata with ancient water harvesting cistern (aljibe) along a transhumance route |
| 7 | San Diego Mine, Rodalquilar | Sierra de Gata with an ancient gold mine, ignimbrites and mine dumps |
| 8 | Rodalquilar Mine, Rodalquilar | Sierra de Gata with an ancient gold mine and gold extraction factory, caldera |
| 9 | La Isleta del Moro | Coastal evolution, alluvial fans and basalt columns |
| 10 | Los Escullos | Fossil dunes, eolianite rock cliff with tafoni (honeycomb weathering) |
| 11 | El Puntal | Pleistocene alluvial fan with dated calcretes (petrocalcic horizon) |
| 12 | Rambla Nogalte, Puerto Lumbreras | Ephemeral river channel, impact of historical flash floods (up to 2500 $m^3/s$) |
| 13 | Lorca Castle, Lorca | Horst and graben site, Lorca-Alhama fault, hogbacks, land use |
| 14 | Puentes Dam, La Parroquia | Impact of massive sedimentation in Puentes reservoir (storage capacity loss) |
| 15 | Rambla Salada, Zarcilla de Ramos | Ephemeral channel and exposure of gypsum-rich and (Quaternary) valley-fill deposits, present-day channel bank failures and bank gullies, gully erosion control using check dams |
| 16 | Sierra de la Torrecilla, La Fuensanta | Impact of land use (almond grove monoculture) on soil erosion by water and tillage, gully erosion control using large check dams and "clear water effects" in downstream gully channel |
| 17 | Rio Alias, Los Alamillos | Evolution of river terrace composition following a river capture |

**Table A1.** *Cont.*

| Geosite Number [a] | Name of the Geosite, and/or Town | Major Features |
|---|---|---|
| 18 | Rambla de los Feos, Los Arejos | River terraces on gypsum and marls |
| 19 | Los Perales | Viewpoint on Sorbas basin, Rio Aguas valley, wind gap (river capture site), large-scale landslides (including rock topple, rockfall and large and deep tension cracks), thick gypsum deposits resting on marls. |
| 20 | Los Molinos | Viewpoint on Sorbas basin, Rio Aguas valley, relief inversion, river capture, gypsum plateau and gypsum karst features (sinkholes and caves) |
| 21 | Lucainena de las Torres | Iron ore mines in Sierra Alhamilla (mica schists) and industrial archaeological site (furnaces to extract iron) |
| 22 | Los Yesos | Undissected Tabernas basin with large-scale irrigated olive monoculture |
| 23 | Los Millares | Viewpoint over dissected Tabernas basin, Rio Andarax and surrounding sierras. Copper-age (Chalcolithic) archaeological site on promontory near the Rio Andarax |
| 24 | Bar Alfaro, Tabernas | Badlands near rambla de Tabernas with exposures of flysch and marls, fault and travertine dam. |
| 25 | Mini Hollywood, Tabernas | Viewpoint on badlands near rambla de Tabernas, Sierra Alhamilla, and on Alfaro hogback. |
| 26 | Rambla Honda, Tabernas | Large-scale fan infilling valley cut into mica schists, hogbacks and remnants of historic settlements with traditional spate irrigation systems (terraces and canals). |
| 27 | La Calahorra castle, La Calahorra | Viewpoint on Sierra Nevada, open-pit iron mine of Marquesada and on Guadix basin. |
| 28 | Esfiliana | Large bank gullies dissecting gently sloping gravelly alluvial fans in the Guadix basin. |
| 29 | Gorafe | Viewpoint over Rio Gor valley cut in Guadix-Baza basin, with large-scale landslides, groundwater calcretes, Bronze-age megalithic park |
| 30 | Alicun de las Torres | Viewpoint over Rio Fardes valley and large travertine dams, hot water springs |
| 31 | Belerda | Viewpoint on Sierra Nevada, Guadix basin infill, large valley-bottom gully and groundwater calcretes |
| 32 | Rio Aguas, Sorbas | Ephemeral stream channel with heterogeneous bedload deposit. |
| 33 | Embalse de Isabel II, Níjar | Valley cut in mica schists of the Sierra de los Filabres, with completely infilled reservoir |
| 34 | El Hoyazo, Níjar | Eroded volcano with volcanic plug and coral reef deposits, ancient garnet mining site |

[a] The geosites were numbered based on the order in which they were visited by the GB-V group.

**Table A2.** Grouping of the 34 geosites into 5 clusters based on features of interest (see the methodology section to understand how the classification was made).

| Geosite Number | Geosite Name | Features of Interest at the Geosite |
|---|---|---|
| 1 | Boca Andarax | Local geo-feature, viewpoint |
| 2 | Torre Garcia | Human–environment interaction, local geo-feature, viewpoint |
| 3 | Las Salinas | Human–environment interaction, local geo-feature, viewpoint |
| 4 | Punta Baja | Local geo-feature, viewpoint |
| 5 | Cerro Pistolas | Human–environment interaction, viewpoint |
| 6 | Albaricoques | Human–environment interaction, viewpoint |
| 7 | San Diego Mine | Human–environment interaction, viewpoint |
| 8 | Rodalquilar Mine | Human–environment interaction, local geo-feature, viewpoint |
| 9 | La Isleta del Moro | Local geo-feature, viewpoint |
| 10 | Los Escullos | Local geo-feature, viewpoint |
| 11 | El Puntal | Local geo-feature |
| 12 | Rambla Nogalte | Human–environment interaction |
| 13 | Lorca Castle | Local geo-feature, viewpoint |
| 14 | Puentes Dam | Human–environment interaction |
| 15 | Zarcilla de Ramos (Rambla Salada) | Local geo-feature |
| 16 | Sierra de la Torrecilla | Human–environment interaction, viewpoint |
| 17 | Rio Alias | Local geo-feature |
| 18 | Rambla de los Feos | Local geo-feature |
| 19 | Los Perales | Local geo-feature, viewpoint |
| 20 | Los Molinos | Local geo-feature, viewpoint |
| 21 | Lucainena de las Torres | Human–environment interaction, viewpoint |

**Table A2.** *Cont*.

| Geosite Number | Geosite Name | Features of Interest at the Geosite |
|---|---|---|
| 22 | Los Yesos | Human–environment interaction, viewpoint |
| 23 | Los Millares | Human–environment interaction, viewpoint |
| 24 | Bar Alfaro | Local geo-feature, viewpoint |
| 25 | Mini Hollywood | Local geo-feature, viewpoint |
| 26 | Rambla Honda | Human–environment interaction, local geo-feature, viewpoint |
| 27 | La Calahorra | Human–environment interaction, viewpoint |
| 28 | Esfiliana | Local geo-feature, viewpoint |
| 29 | Gorafe | Human–environment interaction, viewpoint |
| 30 | Alicun de las Torres | Local geo-feature, viewpoint |
| 31 | Belerda | Local geo-feature, viewpoint |
| 32 | Rio Aguas | Local geo-feature |
| 33 | Embalse de Isabel II | Human–environment interaction, viewpoint |
| 34 | El Hoyazo | Local geo-feature, viewpoint |

**Table A3.** Socio-demographic profile of the three respondent groups (*n* = number of persons).

| | Socio-Demographic Variables | NGB-NV Group (*n* = 104) Frequency *n* (%) | GB-NV Group (*n* = 43) Frequency *n* (%) | GB-V Group (*n* = 29) Frequency *n* (%) | Total Respondents (*n* = 176) Frequency *n* (%) |
|---|---|---|---|---|---|
| *Gender* [a] *(n = 171)* | Female | 51 (49) | 14 (32.6) | 7 (29.2) | 72 (42.1) |
| | Male | 53 (51) | 29 (67.4) | 17 (70.8) | 99 (57.9) |
| *Age (n = 147)* | 18–29 | 46 (44.2) | 25 (58.1) | NA [b] | 71 (48.3) |
| | 30–49 | 25 (24.1) | 10 (23.3) | NA | 35 (23.8) |
| | 50–77 | 33 (31.7) | 8 (18.6) | NA | 41 (27.9) |
| *Education level* [a] *(n = 169)* | Bachelor's degree (*n* = 54) | 34 (34.3) | 11 (25.6) | 9 (33.3) | 54 (32.0) |
| | Master's degree | 58 (58.6) | 24 (55.8) | 16 (59.3) | 98 (58.0) |
| | PhD degree | 7 (7.1) | 8 (18.6) | 2 (7.4) | 17 (10.0) |
| *Country of residence (n = 176)* | Belgium | 67 (64.4) | 29 (67.4) | 28 (96.6) | 125 (71.0) |
| | Other | 37 (35.6) | 14 (32.6) | 1 (3.4) | 52 (29.0) |
| *Continents visited* [a] *(n = 175)* | No travel outside my continent | 25 (24) | 14 (32.5) | 9 (32.1) | 48 (27.4) |
| | Visited one other continent | 41 (39.4) | 7 (16.3) | 6 (21.4) | 54 (30.9) |
| | Visited 2–3 other continents | 21 (20.2) | 15 (34.9) | 8 (28.6) | 44 (25.1) |
| | Visited 4–5 other continents | 17 (16.4) | 7 (16.3) | 5 (17.9) | 29 (16.6) |
| Primarily preferred attraction types (*n* = 173) | Biodiversity | 8 (7.7) | 1 (2.3) | 4 (15.4) | 13 (7.5) |
| | Biodiversity and culture-history | 11 (10.6) | 0 (0.0) | 0 (0.0) | 11 (6.4) |
| | Biodiversity and geology-landscape | 2 (1.9) | 1 (2.3) | 0 (0.0) | 3 (1.7) |
| | Biodiversity, culture- history and geology-landscape | 20 (19.2) | 11 (25.6) | 0 (0.0) | 31 (17.9) |
| | Culture-history | 28 (26.9) | 4 (9.3) | 5 (19.2) | 37 (21.4) |
| | Culture-history and geology-landscape | 17 (16.4) | 10 (23.3) | 0 (0.0) | 27 (15.6) |
| | Geology-landscape | 18 (17.3) | 16 (37.2) | 17 (65.4) | 51 (29.5) |

[a] Missing values for the GB-V group. [b] NA = data not available.

**Table A4.** Comparison of mean scenic beauty ratings (standard deviations in parentheses) of the 34 individual geosites by the three groups of respondents, grouped following their geoscience background and field visits: NGB-NV = No-Geoscience Background-No Visit to geosites; GB-NV = Geoscience Background-No Visit to geosites; GB-V = Geoscience Background-Visit to geosites.

| Geosite Number | Name of the Geosite | Mean of the NGB-NV Group | Mean of the GB-NV Group | Mean of the GB-V Group | F Value |
|---|---|---|---|---|---|
| 1 | Boca Andarax | 3.25 (0.93) | 3.02 (1.01) | 2.28 (0.92) | 11.92 *** |
| 2 | Torre Garcia | 3.04 (0.92) | 2.88 (0.88) | 2.97 (0.82) | 0.46 |
| 3 | Las Salinas | 2.61 (0.84) | 2.63 (1.05) | 3.31 (1) | 6.95 ** |
| 4 | Punta Baja | 3.85 (0.89) | 4.35 (0.61) | 4.79 (0.41) | 19.65 *** |
| 5 | Cerro Pistolas | 3.56 (0.9) | 3.65 (0.9) | 3.10 (1.11) | 3.35 * |
| 6 | Albaricoques | 2.62 (0.96) | 2.65 (1.02) | 2.83 (0.85) | 0.56 |
| 7 | San Diego Mine | 3.5 (0.91) | 3.74 (1.11) | 3.48 (0.95) | 1.06 |
| 8 | Rodalquilar Mine | 2.87 (1.01) | 3.12 (1.1) | 3.86 (0.79) | 11.35 *** |
| 9 | La Isleta del Moro | 4 (0.86) | 3.81 (1.03) | 3.69 (1.04) | 1.51 |
| 10 | Los Escullos | 4.28 (0.85) | 4.58 (0.7) | 4.45 (0.78) | 2.26 |
| 11 | El Puntal | 2.37 (0.89) | 2.05 (1.07) | 1.83 (1.07) | 4.20 * |

**Table A4.** *Cont.*

| Geosite Number | Name of the Geosite | Mean of the NGB-NV Group | Mean of the GB-NV Group | Mean of the GB-V Group | F Value |
|---|---|---|---|---|---|
| 12 | Rambla Nogalte | 2.09 (1.06) | 2.26 (1.14) | 2.04 (0.95) | 0.39 |
| 13 | Lorca Castle | 2.93 (0.98) | 2.86 (1.04) | 3.86 (0.83) | 11.69 *** |
| 14 | Puentes Dam | 2.98 (0.9) | 2.95 (1.17) | 2.45 (1.02) | 3.38 * |
| 15 | Zarcilla de Ramos (Rambla Salada) | 3.54 (0.92) | 3.67 (0.99) | 3.59 (0.68) | 0.34 |
| 16 | Sierra de la Torrecilla | 3.4 (0.96) | 3.49 (1.05) | 3.45 (0.83) | 0.12 |
| 17 | Rio Alias | 2.86 (1) | 3.33 (0.94) | 3.21 (0.9) | 4.15 * |
| 18 | Rambla de los Feos | 3.07 (0.95) | 3.53 (0.93) | 2.69 (0.85) | 7.56 *** |
| 19 | Los Perales | 3.13 (0.98) | 3.65 (1.15) | 3.52 (1.02) | 4.45 * |
| 20 | Los Molinos | 3.44 (0.98) | 3.7 (1.08) | 4.14 (0.92) | 5.69 ** |
| 21 | Lucainena de las Torres | 3.93 (0.85) | 3.86 (1.08) | 3.48 (0.83) | 2.8 |
| 22 | Los Yesos | 3.07 (1.05) | 2.72 (1.05) | 2.07 (0.96) | 10.82 *** |
| 23 | Los Millares | 3.61 (0.95) | 3.6 (1.07) | 3.62 (1.08) | 0.00 |
| 24 | Bar Alfaro | 3.53 (0.99) | 4.21 (0.94) | 4.38 (0.86) | 13.19 *** |
| 25 | Mini Hollywood | 3.76 (1.03) | 4.51 (0.63) | 4 (0.89) | 10.08 *** |
| 26 | Rambla Honda | 3.23 (1.04) | 3.72 (0.91) | 3.79 (1.01) | 5.69 ** |
| 27 | La Calahorra | 3.23 (1.05) | 3.53 (0.98) | 4.21 (0.82) | 10.88 *** |
| 28 | Esfiliana | 3.37 (1.01) | 3.63 (1.05) | 2.76 (0.95) | 6.61 ** |
| 29 | Gorafe | 2.73 (0.93) | 2.67 (0.99) | 4.28 (0.7) | 35.61 *** |
| 30 | Alicun de las Torres | 3.57 (0.97) | 3.88 (0.79) | 4.83 (0.54) | 23.62 *** |
| 31 | Belerda | 3.37 (0.88) | 3.63 (1.16) | 3 (1) | 3.6 * |
| 32 | Rio Aguas | 2.3 (0.95) | 2.49 (0.96) | 2.14 (0.92) | 1.24 |
| 33 | Embalse de Isabel II | 2.89 (1) | 3.07 (0.99) | 3.1 (1.01) | 0.77 |
| 34 | El Hoyazo | 3.5 (1.01) | 3.65 (0.92) | 4.38 (0.56) | 10.10 *** |

\* $p < 0.05$; \*\* $p < 0.01$.; \*\* $p < 0.001$.

**Table A5.** Mean scientific value ratings of the 34 geosites, rated by the GB-V group ($n = 29$ persons).

| Geosite Number | Geosite Name | Scientific Value Rating | Geosite Number | Geosite Name | Scientific Value Rating |
|---|---|---|---|---|---|
| 1 | Boca Andarax | 3.03 | 18 | Rambla de los Feos | 3.28 |
| 2 | Torre Garcia | 3.25 | 19 | Los Perales | 3.72 |
| 3 | Las Salinas | 3.29 | 20 | Los Molinos | 3.89 |
| 4 | Punta Baja | 4.24 | 21 | Lucainena de las Torres | 3.00 |
| 5 | Cerro Pistolas | 3.45 | 22 | Los Yesos | 2.39 |
| 6 | Albaricoques | 2.83 | 23 | Los Millares | 3.72 |
| 7 | San Diego Mine | 3.10 | 24 | Bar Alfaro | 4.52 |
| 8 | Rodalquilar Mine | 3.59 | 25 | Mini Hollywood | 3.86 |
| 9 | La Isleta del Moro | 3.17 | 26 | Rambla Honda | 3.48 |
| 10 | Los Escullos | 4.32 | 27 | La Calahorra | 3.32 |
| 11 | El Puntal | 2.64 | 28 | Esfiliana | 3.29 |
| 12 | Rambla Nogalte | 3.52 | 29 | Gorafe | 4.14 |
| 13 | Lorca Castle | 3.79 | 30 | Alicun de las Torres | 4.59 |
| 14 | Puentes Dam | 2.86 | 31 | Belerda | 3.06 |
| 15 | Zarcilla de Ramos (Rambla Salada) | 3.64 | 32 | Rio Aguas | 2.82 |
| 16 | Sierra de la Torrecilla | 3.63 | 33 | Embalse de Isabel II | 3.62 |
| 17 | Rio Alias | 3.71 | 34 | El Hoyazo | 4.66 |

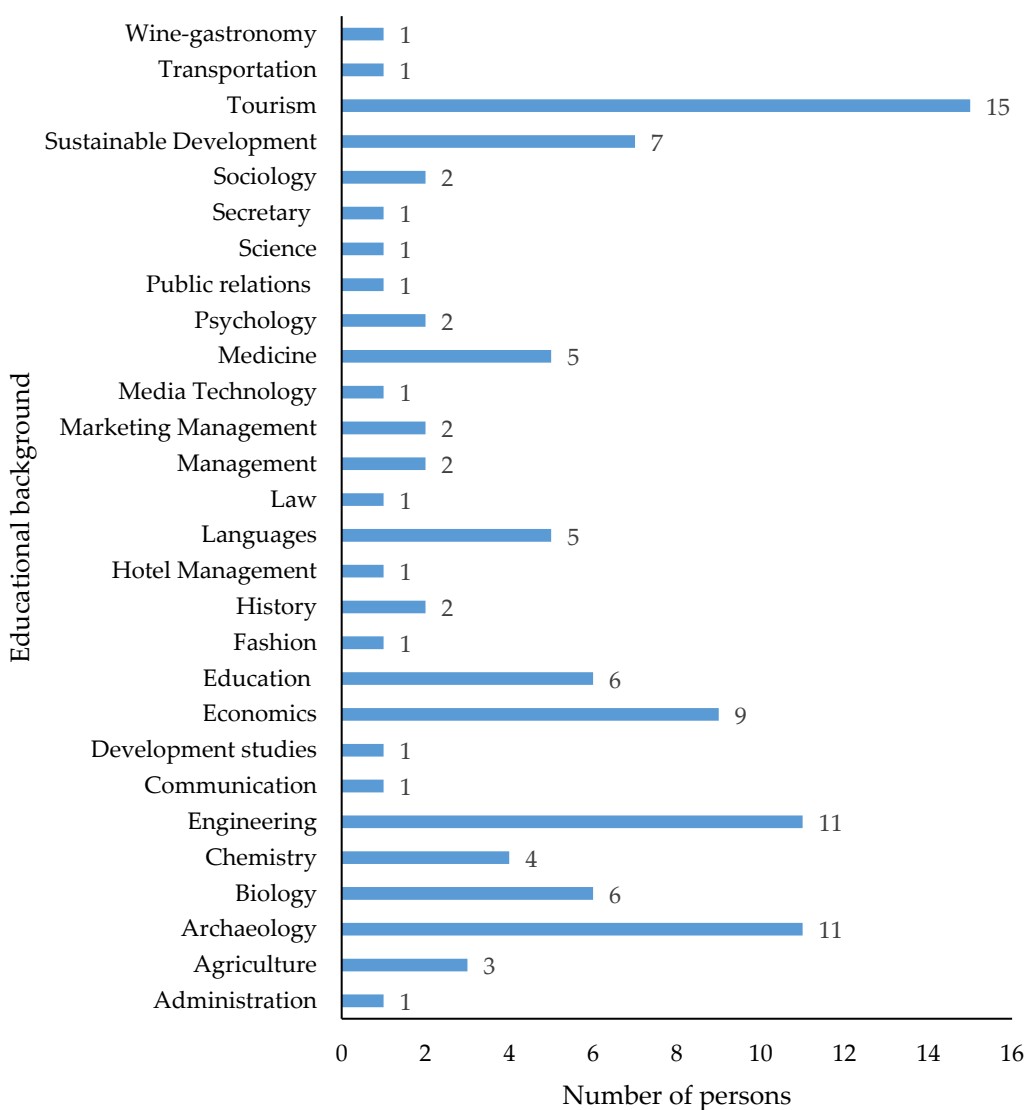

**Figure A1.** Number of persons in the NGB-NV group (total = 104) and their educational background.

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
