# Peer review of "The Scenic Beauty of Geosites and Its Relation to Their Scientific Value and Geoscience Knowledge of Tourists: A Case Study from Southeastern Spain"

_land, doi:10.3390/land10050460_

Round 1
Reviewer 1 Report
The article is very interesting, well structured and clearly written. It addresses the relationship between (geoscience) knowledge and the aesthetic perception of geo-localities.
The research has a very precise methodology. This is very well described in the article.
Nevertheless, I have a few remarks on the article:
The authors do not take into account the theory of photography and image perception - composition by man. The perception of the observer can be influenced, for example, by the height of the horizon in the photograph, the shape of the photograph (square vs. wide-angle). The authors cannot influence this aspect retrospectively, but should mention it in the discussion. There should also be a systemic approach to photographs when used to detect human perception.
Research could not provide answers to hypothesis, despite a precise overall approach to methodology. The reason is the small variability of localities as well as the definition of "scientific value of geosites". Scientific value may not always be visible at first glance.
The results also cannot be generalized. De facto, this is the respondents' attitude to the nature of the landscape in one region and therefore cannot be generalized.
How are the results affected by the choice of respondents, resp. their thematic focus? As the authors rightly state, this is not only a statistically insignificant sample of the addressed respondents, but the perception of a person is influenced not only by the achieved education, but by the overall subjective approach. E.g. Different target groups of tourists / visitors have different perceptions of aesthetics. This aspect is insufficiently mentioned in the article. How was it taken into account when creating the set of respondents?
In summary, these are interesting results, which, however, are difficult to interpret correctly both due to errors in working with the photographic presentation of sites, in choosing a set of respondents and in choosing model sites.
Both the article and the research are still interesting. The authors just have to rethink the interpretation of the results.
Author Response
Dear Editor and Reviewer 1,
We thank you for your positive feedback. We are grateful for the valuable and constructive comments which helped us to further improve the paper. In this reply, comments of Reviewer 1 are typed in normal font with black color, while our answers to the comments are typed in red. Paragraphs/statements copied from the revised manuscript in response to the Reviewer’s comments are also included here for clarity (and typed in red + italic). In addition, “Line numbers” are also cited here for your convenience to refer where the added statements/paragraphs are in the revised manuscript.
Getaneh Addis Tessema, on behalf of all authors

Reviewer 2 Report
Dear Authors,
It was a delight to read your well-written and well-documented paper. Therefore, I have no further observations and in my opinion, it should be published as it is.
This is a very interesting paper, well-written and documented; which presents the relation between knowledge and aesthetic perception of geosites. At this stage, I would recommend some Minor Revisions: The number of references is really high; at a first glance, out of 110 references, almost only 75-80 are really necessary. In your review, please include the following reference, which is highly relevant for your paper: https://doi.org/10.1007/s12371-017-0252-1 and discuss it within the context of your paper. An important aspect of your paper is the use of the term "geosites" and not "geomorphosites" as it is in the literature. If you have no real reason to do so, please replace for the entire manuscript with "geomorphosites".
Kind regards.
Author Response
Dear Editor and Reviewer 2,
We thank you for your positive feedback. We are grateful for the valuable and constructive comments which helped us to further improve the paper. In this reply, comments of Reviewer 2 are typed in normal font with black color, while our answers to the comments are typed in red. Paragraphs/statements copied from the revised manuscript in response to the Reviewer’s comments are also included here for clarity (and typed in red + italic). In addition, “Line numbers” are also cited here for your convenience to refer where the added statements/paragraphs are in the revised manuscript.
Getaneh Addis Tessema, on behalf of all authors
